# Surgical Conservative Management of a Retained Placenta after Angular Pregnancy, a Case Report and Literature Review

**DOI:** 10.3390/diagnostics13233492

**Published:** 2023-11-21

**Authors:** Giovanna Bitonti, Paola Quaresima, Giampiero Russo, Costantino Di Carlo, Giuseppina Amendola, Rosanna Mazzulla, Roberta Venturella, Michele Morelli

**Affiliations:** 1Department of Obstetrics and Gynecology, Magna Graecia University, Viale Europa, 87100 Catanzaro, Italy; giovanna.bitonti@gmail.com (G.B.); venturella@unicz.it (R.V.); 2Consultorio Familiare, Azienda Sanitaria Provinciale Cosenza, Centro Sanitario, Ponte Pietro Bucci, Arcavacata, 87036 Cosenza, Italy; giampierorusso@tiscali.it; 3Department of Public Health, University of Naples Federico II, Via Pansini 5, 80127 Naples, Italy; cdicarlo@unina.it; 4Department of Obstetrics and Gynecology, Azienda Ospedaliera di Cosenza SS Annunziata, Via Felice Migliori 1, 87100 Cosenza, Italymorellimichele122@gmail.com (M.M.)

**Keywords:** angular pregnancy, intrauterine growth restriction, retained placenta, hysterotomy

## Abstract

Angular pregnancies are rare and difficult to diagnose. Evidence suggests they are associated with a higher risk of intrauterine growth restriction and abnormal third stage of labor due to a retained placenta. The lack of standardized AP diagnostic criteria impacts on their correct identification and makes the treatment of potential complications challenging. We present a case of the successful conservative surgical management of a retained placenta after a term AP also complicated by intrauterine growth restriction. Moreover, to identify the best evidence regarding AP diagnostic criteria and retained placenta therapeutic approaches, we have realized an expert literature review.

## 1. Introduction

Angular pregnancy (AP) is a rare entity with less than 100 reported cases in the available literature [1,2]. Howard Kelly first described it in 1898 as an implantation of the embryo in the lateral angle of the uterine cavity, medial to the utero-tubal junction [3]. It must be differentiated from interstitial pregnancy by the different displacement of the round ligament, which, in turn, is lateral in AP, and medial in the interstitial ones [4].

However, this theoretical definition appears to be challenging in routine clinical practice [5]. According to Howard Kelly’s definition [2,6,7], AP can be considered as potentially evolutive, and pregnancy outcomes are good [8]. On the other hand, interstitial pregnancy is more often associated with serious complications such as uterine rupture [9]. Therefore, the correct differential diagnosis is crucial.

AP can present as an asymmetrical and painful enlargement of the uterus during the first trimester of pregnancy [8]. If suspected in advanced gestation, anamnesis, ultrasound, the location and characteristics of the placenta, which appears thickened and confined asymmetrically in the uterine angle, are crucial signs for the identification of such cases [3].

Term AP appears to be associated with an increased risk of intrauterine growth restriction and abnormal third stage of labor due to a retained placenta [10]. The lack of standardized AP diagnostic criteria impacts on their correct identification and makes the treatment of potential complications challenging. We present a case of the successful conservative surgical management of a retained placenta after a term AP also complicated by intrauterine growth restriction. Moreover, to identify the best evidence regarding AP diagnostic criteria and retained placenta therapeutic approaches, we have realized an expert literature review.

## 2. Case Report

A 37-year-old nulliparous woman was admitted to the obstetrics and gynecology unit of the Annunziata hospital (Cosenza, Italy), with a diagnosis of premature rupture of membranes (PROMs), at 36 weeks and 3 days of gestation. The maternal anamnesis was negative and pregnancy conception spontaneous. During the first trimester dating ultrasound, a suspicion of an abnormal embryo implantation in the upper right angle of the uterus arose. The gestational sac was fully surrounded by endometrium, without an “interstitial line sign” and with a myometrial mantle thickness of 9 mm; the whole picture was highly suggestive of an angular pregnancy. After an accurate consultation, the woman opted for an expectant management approach and close follow-up.

Pregnancy was complicated by a late-onset intrauterine growth restriction according to the most recent diagnostic criteria of the International Society of Ultrasound in Obstetrics and Gynecology (ISUOG) [11] (estimated fetal weight below the 3rd centile with abnormal uterine arteries Doppler). At hospital admission upon ultrasound review, the placenta appeared to be located in the upper right lateral side of the uterus (Figure 1).

A vaginal delivery of a 2 kg male newborn occurred spontaneously after two days. The Apgar score at the 1st, 5th and 10th minute was, respectively, 10/10/10. Thirty minutes after delivery, despite active management of the third stage of labor, the placenta was still not delivered. An ultrasound evaluation showed the presence of an entangled placenta within the right uterine angle, and, according to Herman Ultrasound findings [12], the myometrium appeared thin at the placenta insertion but more than 2 mm (Figure 2), and there was no evidence of accreta spectrum.

Thus, after one hour of observation and the acquisition of a detailed informed consent, mentioning conservative versus non-conservative potential treatment, the patient was conducted in the operating room with the diagnosis of a retained placenta after angular pregnancy.

Two consecutive manual removals were attempted, without success. A third one by a senior operator was performed and failed due to a mechanical difficulty at placental extraction. Therefore, given the persistent uterine bleeding, around 600 mL, the administration of Oxytocin (20 IU, 500 mL normal saline, continuous infusion) and Tranexamic acid (1 g, 100 mg/mL) was started. The patient was informed about the need for a surgical treatment; therefore, she underwent a laparotomic exploration. The uterus appeared asymmetrically enlarged, with an irregular right cornual bulge, at the site of the trapped placenta (Figure 3).

Given the good hemodynamic control, taking into account the young maternal age and the desire for fertility sparing, a hysterotomy of the right cornual region was performed with the subsequent manual removal of the entangled placenta. As a conclusion, the uterine breach was sutured from the posterior to anterior uterine wall with a single layer of interrupted stitches (Figure 4A–D). The total estimated blood loss was 800 mL, with a post-operative Hb of 9.3 g/dL.

No intra-operative complication occurred. The post-partum was uneventful and after five days, the patient was discharged. To date, the woman is in a good health condition.

Placental histopathological analysis has shown areas of placental infarction characterized by ischemic and fibrinoid necrosis, images available on Figure 5.

## 3. Literature Review

### 3.1. Search Strategy

The PubMed, Scopus, Google Scholar and Web Of Science databases were searched electronically in February 2023, considering only papers from 1930 to the current year. Given the too heterogeneous definition of angular, interstitial and cornual pregnancies, it was decided to use the following three different search strings: “angular pregnancy AND retained placenta”, “cornual pregnancy AND retained placenta” and “Interstitial pregnancy AND retained placenta”. The reference lists of relevant articles and reviews were hand-searched for additional reports. The inclusion criteria were women diagnosed with an AP, that ended up with a live birth newborn, in which a retained placenta occurred. We excluded all the cases in which angular pregnancies ended too early, were terminated in the first trimester, cases of abnormally adherent placenta (placenta accreta) and cases finally diagnosed as interstitial or cornual pregnancies. The different AP therapeutic approaches were evaluated: surgical (demolitive and conservative) and/or conservative (pharmacological, manual removals and expectant management). For each kind of management, every complication was described. We decided to consider only full-text articles published in English. Two authors (G.B., P.Q.) independently searched and reviewed all papers and agreed about potential eligibility or paucity. Afterwards, they extracted the relevant data from each paper.

### 3.2. Literature Review Results

Between the different search strings and databases, a total of 81 papers were identified. No guidelines or other systematic reviews were found. Forty-six papers were excluded because of duplication (41), the full text not being available (2) and not being in the English language (3). Thirty-five papers were therefore assessed for a first analysis: twenty-three were excluded because they did not meet the eligibility criteria. Twelve studies were included in the final analysis involving fifteen cases [3,10,13,14,15,16,17,18,19,20,21,22], as shown in the Prisma flow diagram available on Figure 6.

All selected cases were singleton pregnancies. The mean maternal age was 27 years old, the gestational age at birth was available for 11/15 (73.3%) cases, the mean gestational age was 35 weeks, the birth weight was described in 8/15 cases and the mean was 2225 g. Based on the available data, an intrauterine growth restriction was found in 50% of the available cases (4/8). Twelve women delivered vaginally (12/15, 80%), whereas three patients underwent a caesarean section (3/15, 20%). In two cases, a maternal history of a uterine Mullerian anomaly was detected (complete and partial septate uterus) and four out of fifteen patients (26.6%) reported having undergone a previous uterine surgery. The presence of a fetal anomaly was reported in three cases out of fifteen (3/15, 20%). In detail, these were a fetal bilateral hydronephrosis associated with an atrial septal defect, a pulmonary agenesis of the right lung and a tetralogy of Fallot [23,24]. The suspicion of being in front of an angular pregnancy during the first trimester dating ultrasound occurred only in one case (1/15, 6.6%), the remaining were diagnosed after birth. One patient underwent post-partum magnetic resonance imaging and computed tomography for a better evaluation of the placental entrapment. We have evaluated the therapeutic approaches for each of the above-mentioned cases; two out of fifteen cases (2/15, 13.3%) were resolved with an expectant management, two (2/15, 13.3%) with a manual removal of the placenta, in one case a curettage allowed for the expulsion of the retained placenta, six women (6/15, 40%) received a conservative surgical treatment such as a hysterotomy and finally, four (4/15, 26.6%) underwent a total or subtotal hysterectomy. The complication rate for the surgical approaches (hysterotomy, hysterectomy, curettage) was 0%, while in the conservatively managed cases (expectant management and manual removal), both fever and hemorrhage requiring blood transfusions occurred in two of four cases (50%). In the two cases complicated with hemorrhage, one patient needed the apposition of compressive uterine sutures, while a uterine artery embolization procedure was required for the other one. Placental histopathological analysis was performed in only one case and no significant abnormalities were reported.

A description of each case is provided on Table 1.

## 4. Discussion

### 4.1. Angular Pregnancy Diagnosis

AP is a difficult entity to diagnose, due to the lack of agreement regarding the definition and differential diagnosis. The words cornual, interstitial and angular pregnancies are often used interchangeably. Cornual pregnancy is a completely different entity, subsequent to a Mullerian anomaly. Indeed, according to its most recent definition, it is considered as “a conception that develops in the rudimentary horn of a uterus with a Mullerian anomaly” [8,23]; the differential diagnosis between angular and interstitial pregnancies is more difficult. Indeed, it is made up on the basis of different parameters: firstly, the direction of the round ligament displacement from the gestational sac, the AP displaces it upward and outward whereas the interstitial one displaces this last medially [8,23,25]. Moreover, in an AP, the embryo locates in the lateral wall of the uterus and is surrounded by the endometrial layer, whereas in the case of an interstitial pregnancy, the embryo locates in the muscular layer of the emerging uterine tube, and it does not take any contact with the endometrial layer [3]. Additional sonographic signs typical of AP are the demonstration of a so-called “double sac” (the presence of surrounding endometrium around the gestational sac, specifically composed of a decidual reaction layer and a chorionic ring) [26,27], a myometrial coat thickness of at least 5 to 8 mm [1] and the absence of the “interstitial line sign” (which is typical of an interstitial pregnancy) [28]. The best time to properly diagnose an AP is the first trimester, during the first trimester dating scan. Doubtful cases may also benefit from a three-dimensional ultrasound evaluation which allows for a better visualization of the endometrial canal, and/or from a magnetic resonance imaging examination. This last technique helps in the differential diagnosis between an AP and interstitial pregnancy; the first may present with a gestational sac surrounded by a T2 hyperintense *endometrium* whereas the second one may present with a gestational sac surrounded by a T2 hypointense *myometrium* [29,30].

It is crucial to make a differential diagnosis between these two entities because they have a completely different prognosis; if angular pregnancy has been defined by the European Society of Human Reproduction and Embryology [26] as “a variation of a normally implanted intrauterine pregnancy, rather than a form of ectopic pregnancy” [26], which associates with a live birth rate in 69–80% cases [2,7], the interstitial one is considered to be an ectopic pregnancy, responsible for around 20% of all maternal deaths [8]. A misdiagnosis of an interstitial pregnancy can lead to uterine rupture and maternal shock early on during pregnancy [9].

### 4.2. AP Complications

#### 4.2.1. Fetal Growth Restriction and Retained Placenta

Fetal growth restriction occurred in four out of eight of the described cases, and none of those women suffered from a Mullerian anomaly, a well-known condition associated with a higher risk of growth restriction [31]. Contrary to the normal placental growth pattern, in the case of an AP, the placenta must adopt a rigid uterine angle shape. This condition, in association with placental adhesion anomalies and muscular weakness at the level of placental plat, has been postulated to play a role in fetal growth [3]. Moreover, recent evidence suggests an underlying association between growth restriction, pre-eclampsia, preterm birth and retained placenta, which reflects the possible impact of a defective placentation on the fetal well-being. All these conditions are part of a larger spectrum of disorders of defective or impaired deep placentation. These disorders are known to be associated with enhanced oxidative stress and apoptosis within the placenta, which in turn have been shown to associate with a retained placenta [32,33]. However, the little numbers of our literature review cases do not allow for any generalization regarding a retained placenta after an AP and growth restriction; therefore, further prospective studies are needed.

#### 4.2.2. Retained Placenta

A higher risk of an abnormal third stage of labor due to a retained placenta in the case of an AP has been reported in the available literature. The incidence is around double with respect to the general obstetric population (4% versus 2.7% in high-income countries and 1.5% in low-income countries) [34,35]. Taking into account that the main factor responsible for placenta delivery after birth is the retro-placental myometrial contraction, it can be postulated that in the presence of an AP, the thin retro-placental myometrial wall causes an ineffective contractility, which may contribute to the occurrence of a retained placenta [36,37]. According to the available guidelines, a prolonged third stage of labor can be diagnosed 30 min after birth if managed actively whereas after 60 min if managed physiologically [37,38]. After birth and prior to diagnosing the placenta to be retained, active management, with Oxytocin administration (10 units, intravenous or intramuscular), uterine massage and umbilical cord gentle traction, is recommended to facilitate a spontaneous placental separation [37,38]. Once a retained placenta is diagnosed, an attempt at manual removal, with an adequate analgesia [37,38,39], should be offered to the patient. If this last technique fails, either a “banjo” curette or large oval forceps (Sopher or Bierer) can be used [34]. However, in the presence of an inaccessible placenta like in the case of an AP, manual removal, but also curettage can be challenging and ineffective. Therefore, in the suspicion of such a case after the failure of conservative treatments, a surgical conservative management, like a hysterotomy at the level of the myometrium overlying the placenta, can be taken into account. Our case demonstrated a retained placenta after an AP had been properly suspected during the first trimester dating scan; therefore, operators’ awareness regarding the potential risk of an abnormal third stage of labor allowed for a successful surgical conservative management without any complications. Up to 2023, only six previous cases have described a conservative surgical management of a retained placenta after an AP; therefore, further studies are needed to better evaluate the efficacy and safety of such a technique.

## 5. Conclusions

AP is a difficult entity to diagnose, but an accurate study of the placenta location allows operators to be aware of being in front of such a condition that in most cases, led to an uneventful pregnancy. In the presence of an AP, attention should be paid to fetal growth assessment through gestation, and, in the eventuality of a retained placenta after birth, if conservative strategies fail, then surgical treatment should be taken into account. With this purpose, a uterine hysterotomy should be considered due to its safety and effectiveness. In such cases, the possible complications associated with the presence of a uterine scar (placenta accreta spectrum disorders and/or uterine rupture) should be taken into account for further pregnancies.

## Figures and Tables

**Figure 1 diagnostics-13-03492-f001:**
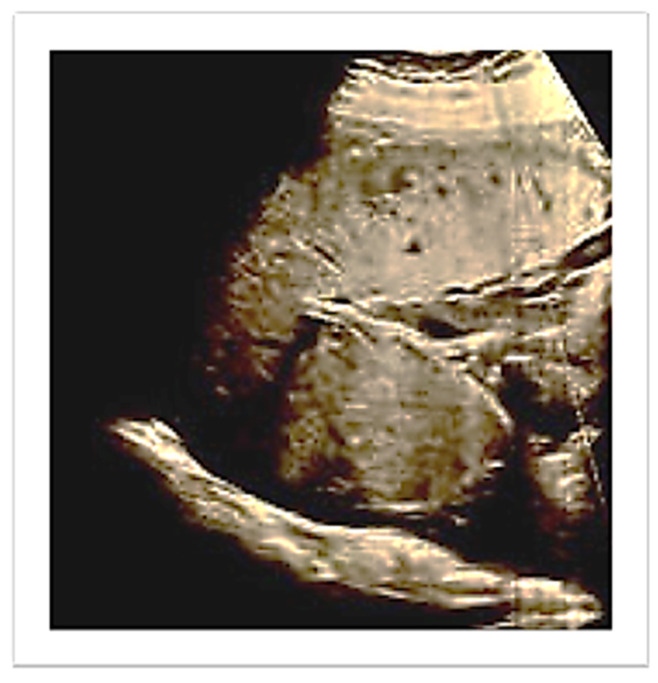
Ultrasound evaluation of placenta location.

**Figure 2 diagnostics-13-03492-f002:**
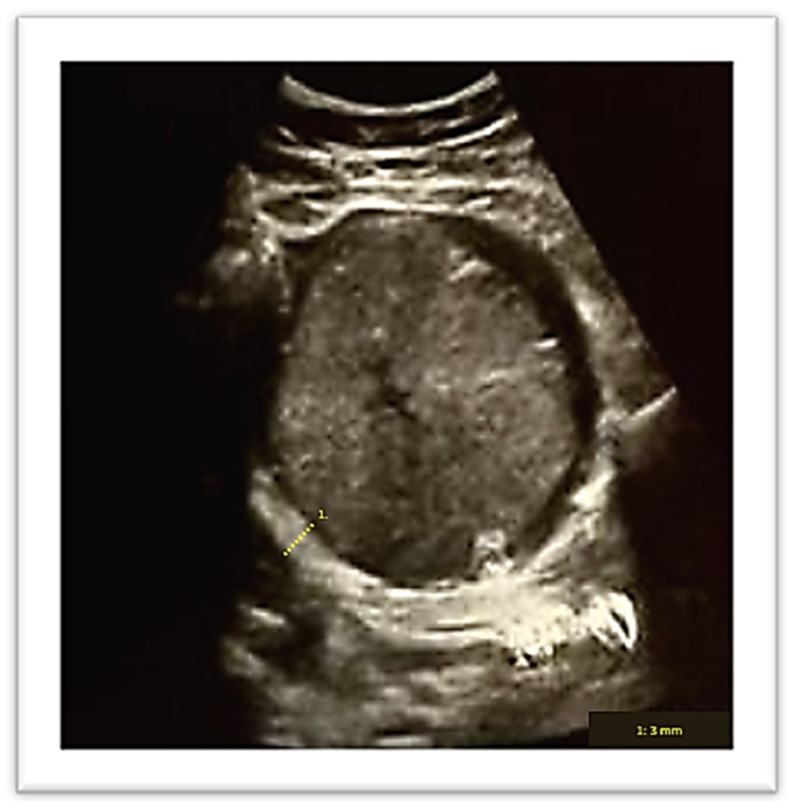
Ultrasound evaluation of placenta location after birth. Number 1 demonstrates myometrial thickness measurement.

**Figure 3 diagnostics-13-03492-f003:**
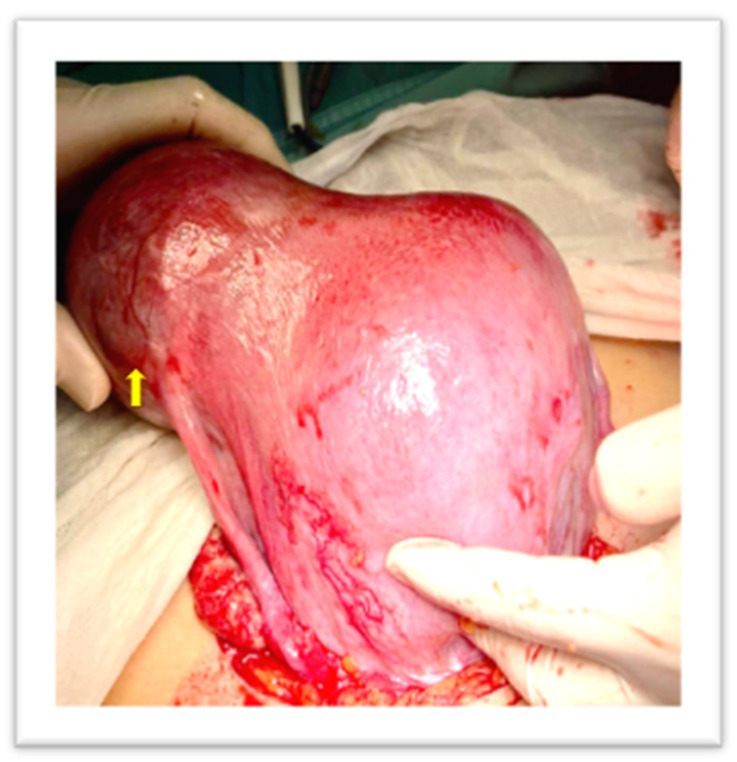
Arrow: Uterine irregular bulge, site of the trapped placenta.

**Figure 4 diagnostics-13-03492-f004:**
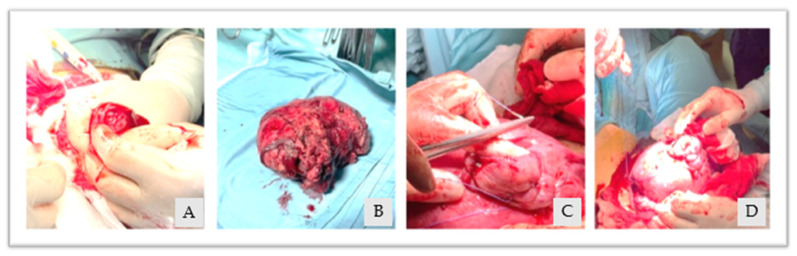
(**A**): uterine hysterotomy, (**B**): removed placenta, (**C**,**D**): uterine breach suture.

**Figure 5 diagnostics-13-03492-f005:**
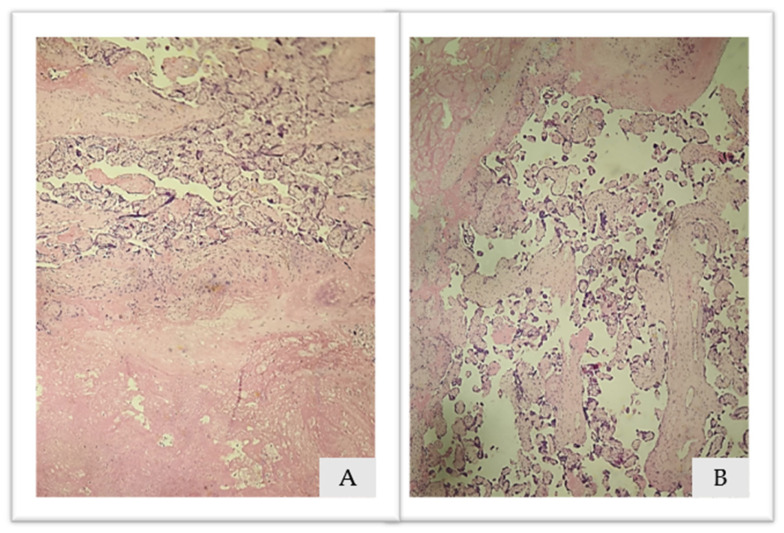
(**A**,**B**). Placental histopathological analysis: Hematoxylin and eosin 10×. (**A**): ischemic necrosis and (**B**): fibrinoid necrosis.

**Figure 6 diagnostics-13-03492-f006:**
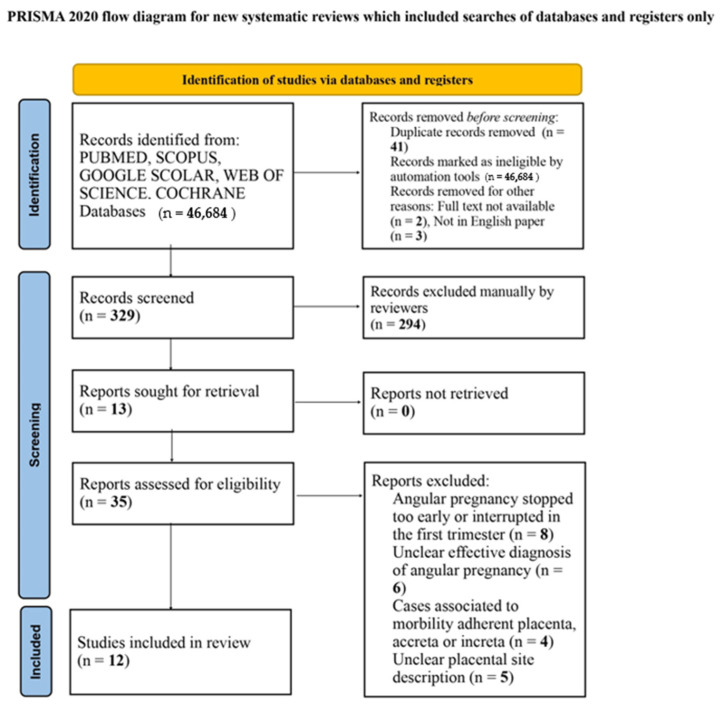
Prisma flow diagram.

**Table 1 diagnostics-13-03492-t001:** AP retained placenta cases, literature review.

	Maternal Age (Years)	Gestational Age (Weeks)	Birth Weight (Grams)	Birth Weight Percentile (Centile)	Maternal or Fetal Pathological Condition	Previous Uterine Surgery	First Trimester Ultrasound Evaluation	Additional Instrumental Evaluations	Mode of Delivery	Therapeutic Approach	Complications
**Gibberd 1936** [13]	25	36	1820	<2.5°	no	no	no	no	vaginal delivery	subtotal hysterectomy	no
**Blaikley 1936, case 1** [14]	n.a	n.a.	n.a.	n.a.	n.a.	n.a.	n.a.	no	vaginal delivery	subtotal hysterectomy	no
**Blaikley 1936, case 2** [14]	n.a.	n.a.	n.a.	n.a.	n.a.	n.a.	n.a.	no	vaginal delivery	subtotal hysterectomy	no
**Naidu 1962** [15]	35	n.a.	n.a.	n.a.	n.a.	no	no	no	vaginal delivery	total hysterectomy	no
**Deckers 2000** [16]	33	38	2685	9°	no	Cesarean section	no	no	vaginal delivery	hysterotomy	no
**Shekhar 2010** [17]	26	30	n.a.	n.a.	n.a.	LPS left adnexectomy, right cystectomy	no	no	vaginal delivery	hysterotomy	no
**Lee 2012** [18]	34	35	2270	19.5°	septate uterus	no	no	no	vaginal delivery	hysterotomy	no
**Amin 2014** [19]	25	31	1800	67.6°	bilateral hydronephrosis, CHD (asd with left to right shunt)	no	no	no	vaginal delivery	hysterotomy	no
**Alanbay 2016** [3]	34	32	1650	6°	no	Cesarean section	yes	no	Cesarean delivery	manual removal	excessive bleeding requiring compression sutures and blood transfusions
**Bijurström 2018** [20]	33	n.a.	n.a.	n.a.	partial septate uterus	curettage	no	no	vaginal delivery	curettage	no
**Xavier 2019** [21]	16	35	n.a.	n.a.	n.a.	n.a.	no	no	vaginal delivery	hysterotomy	no
**Nakatsuka 2020, case 1** [10]	30	39	3212	32.2°	no	no	no	CT, MRI	vaginal delivery	expectant management	fever
**Nakatsuka 2020, case 2** [10]	31	36	1810	<2.5°	pulmonary agenesis of the fetal left lung	no	no	no	Cesarean delivery	expectant management	fever, haemorrhage requiring UAE and blood transfusions
**Nakatsuka 2020, case 3** [10]	31	37	2550	10.9°	tetralogy of Fallot	no	no	no	Cesarean delivery	manual removal	No
**Rajbhar 2021** [22]	26	34	n.a.	n.a.	no	no	no	no	vaginal delivery	hysterotomy	no

Legend: n.a: not available; LPS: laparoscopic; CHD: congenital heart defect; CT: computed tomography; MRI: magnetic resonance imaging; UAE: uterine arteries embolization.

## Data Availability

Data are contained within the article.

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
