# Peer review of "Surgical Conservative Management of a Retained Placenta after Angular Pregnancy, a Case Report and Literature Review"

_diagnostics, 2023, doi:10.3390/diagnostics13233492_

Round 1
Reviewer 1 Report
Comments and Suggestions for Authors
The Authors present an unusual case of a retained placenta after
angular pregnancy associated with literature review. However, the literature search is not complete as they don't include the Web Of Science database. The literature review usually include PubMed, Scopus, and WOS. Overall, it seems the Authors performed a systematic review, therefore they should follow and adequately present the PRISMA flow diagram.
Was the placenta of their case sent for histological analysis? If so, it should be interesting to mention the histological findings. The fetus presented IUGR, was there any correlation between IUGR and placental findings (if analyzed)?
The Authors describe 15 literature cases. Was the placenta analyzed in any of them? Was there anything unusual worth of mentioning?
Comments on the Quality of English LanguageThe manuscript requires accurate revision.
Author Response
Reviewer 1
Dear reviewer thanks for the precious suggestions. We are going to reply to each comment.
- angular pregnancy associated with literature review. However, the literature search is not complete as they don't include the Web Of Science database. The literature review usually include PubMed, Scopus, and WOS. Overall, it seems the Authors performed a systematic review, therefore they should follow and adequately present the PRISMA flow diagram.
R: Dear reviewer Web Of Science database has been also included in the literature review, there was a typing error and as a mistake it has not been mentioned within the methods. We are going to correct the manuscript and also include PRIMA flow diagram.
Line 117 “Pubmed, Scopus, Google Scholar and Web Of Science database databases were searched”
Figure 6 with Prisma Flow Diagram has been added and it was reported on line 140.
- Was the placenta of their case sent for histological analysis? If so, it should be interesting to mention the histological findings.
R: Dear reviewer, yes, the placenta has been sent for histological analysis, images and description have been added on Lines 107,108: “Placental histopathological analysis has shown: areas of placental infarction characterized by ischemic and fibrinoid necrosis, images available on Figure 5.” Fig.5 has been added
- The fetus presented IUGR, was there any correlation between IUGR and placental findings (if analyzed)?
R: Dear reviewer, yes, the placenta histology was compatible with usual findings of placental insufficiency. A description of placental histopathological findings has been added on lines 107,108.
- The Authors describe 15 literature cases. Was the placenta analyzed in any of them? Was there anything unusual worth of mentioning?
R: Dear reviewer placenta analysis has been performed in only one case and no abnormalities were reported. Details added on lines 168,169

Reviewer 2 Report
Comments and Suggestions for Authors
your study is about a surgical management of an incarcerated placenta, a last resort intervention after a succesful vaginal delivery.
congratulations for safely management of the case however there are several issues to be addressed
- if manual removal failed than please specify the reasons - if it was about a uterine contraction which trapped the placenta than did you use miorelaxants such as nitroglicerine?
- a clear hp result image should be added in order to exclude PAS as a cause for trapped placenta
- regarding follow up in conclusion section a clear implication of your intervention should also be added - possible scar interraction with intramyometrial part of fallopian tube causing obstruction and also high risk of uterine rupture in a subsequent pregnancy. Also the risk of a PAS associated with the scar.
- a take home message with a clear recommendation to avoid termination of AP in the first trimester even if such situations could occur.
Author Response
Reviewer 2
Dear reviewer thanks for the precious suggestions. We are going to reply to each comment.
- your study is about a surgical management of an incarcerated placenta, a last resort intervention after a succesful vaginal delivery. congratulations for safely management of the case however there are several issues to be addressed
R: Dear reviewer thanks for the kind words.
- if manual removal failed than please specify the reasons - if it was about a uterine contraction which trapped the placenta than did you use miorelaxants such as nitroglicerine?
R: Manual removal failed due to a mechanical difficulty of the placenta to be delivered. A specification added on Lines 84,85. “Two consecutive manual removals were attempted, without success. A third one by a senior operator was performed, and failed due to a mechanical difficulty at placental extraction.”
Moreover, Nitroglicerine has not been offered to the woman, this last treatment is not a medical option by our national guidelines in case of retained placenta.
- a clear hp result image should be added in order to exclude PAS as a cause for trapped placenta
R: Dear reviewer, unfortunately the US machine used to perform the postnatal evaluation of the retained placenta was the labor ward one, only used for emergencies. Therefore, image quality was not excellent. However, none of the usual signs for placenta accreta were present in our case. None: loss of the normal hypoechoic plane in the myometrium beneath the placental bed and/or multiple placental lacunae and/or loss of the normal hyperechoic line separating the urinary bladder wall from the uterus and/or thinning of the myometrium less to 1 mm.
Figure 2 aims to demonstrate the absence of usual signs for placenta accreta spectrum and the myometrial thickness, thin, but not less than 1 mm. A specification has been added on Lines 72-74 “the myometrium appeared thin at placenta insertion but more than 2 mm (Figure 2), there was no evidence of accreta spectrum”. Figure 2 has been updated with the measurement of myometrial thickness.
- regarding follow up in conclusion section a clear implication of your intervention should also be added - possible scar interraction with intramyometrial part of fallopian tube causing obstruction and also high risk of uterine rupture in a subsequent pregnancy. Also the risk of a PAS associated with the scar.
R: Dear reviewer thanks for your precious suggestion, a sentence has been added accordingly on lines 259-261 “In such cases, the possible complications associated to the presence of a uterine scar (placenta accreta spectrum disorders and or uterine rupture) should be taken into account for further pregnancies.”
- a take home message with a clear recommendation to avoid termination of AP in the first trimester even if such situations could occur.
R: Dear reviewer the following sentence has been added on lines 253-255 “AP is a difficult entity to diagnose, but an accurate study of placenta location allows operators to be aware of being in front of such condition that in most cases led to an uneventful pregnancy”

Reviewer 3 Report
Comments and Suggestions for Authors
This case report describes the outcomes of a retained placenta in a patient with possibly an angular pregnancy. The case report itself is briefly presented and the author discusses the outcome of these cases with a narrative literature review.
The manuscript is well structured, and a pleasure to read. Overall, the topic is relevant, and deserves to be published. Minor comments that have to be addressed:
Specific comments:
1. Line 58: “watchful waiting”.. rather use the term ‘expectant management approach’
2. Line 62: “the placenta resulted” .. rather use state ‘ upon ultrasound review the placenta appeared to be…’
3. Line 68: Was the woman induced or did labour ensue spontaneously?
4. Line 73: What is meant by the myometrium appeared thin? Could you state the measurement?
5. Figure 2: Please highlight/indicate the myometrium and where the measurement was taken.
6. Line 78: Please state the medical management that was given. Was the patient stable? What was the estimated blood loss before surgical management?
7. Line 82: Beyond manual removal what other surgical strategies was used to try and remove the placenta?
8. Line 83: What was the condition of the patient at this stage, i.e., haemodynamically?
9. Line 100: What was the total estimated blood loss? What was her haemoglobin postoperatively? Was any blood transfusion done?
Author Response
Reviewer 3:
Dear reviewer thanks for the precious suggestions and kind words for our manuscript. We are going to reply to each comment.
This case report describes the outcomes of a retained placenta in a patient with possibly an angular pregnancy. The case report itself is briefly presented and the author discusses the outcome of these cases with a narrative literature review.
The manuscript is well structured, and a pleasure to read. Overall, the topic is relevant, and deserves to be published. Minor comments that have to be addressed:
Specific comments:
- Line 58: “watchful waiting”.. rather use the term ‘expectant management approach’
R: dear reviewer thanks for the suggestion the sentence has been modified accordingly.
- Line 62: “the placenta resulted” .. rather use state ‘ upon ultrasound review the placenta appeared to be…’
R: dear reviewer thanks for the suggestion the sentence has been modified accordingly.
- Line 68: Was the woman induced or did labour ensue spontaneously?
R: The patient laboured spontaneously. A specification added on line 68
- Line 73: What is meant by the myometrium appeared thin? Could you state the measurement?
R: Dear reviewer details regarding myometrial thickness have been added on line 73 “the myometrium appeared thin at placenta insertion but more than 2 mm (Figure 2).”
- Figure 2: Please highlight/indicate the myometrium and where the measurement was taken.
R: Dear reviewer a measure of myometrial thickness has been performed and Figure 2 updated.
- Line 78: Please state the medical management that was given. Was the patient stable? What was the estimated blood loss before surgical management?
R: The patient received an active management of the third stage of labor (line 70), details about medical treatment have been added on lines 84-87 “the administration of Oxytocin (20 IU, 500 ml normal saline, continuous infusion) and Tranexamic acid (1 g, 100 mg/ml) has been started “.
The woman has been hemodynamically stable through the entire postpartum period of time up to the end of the surgical procedure, as specified on line 94, before surgical management the blood loss was estimated to be around 600 ml (detail added on line 85)
- Line 82: Beyond manual removal what other surgical strategies was used to try and remove the placenta?
R: An active management of the third stage of labor was performed with the aim to avoid abnormal third stage of labor, no other strategies have been employed other than the three manual removal attempts described within the manuscript.
- Line 83: What was the condition of the patient at this stage, i.e., haemodynamically?
R: the patient has always been haemodynamically stable, as specified on line 94
- Line 100: What was the total estimated blood loss? What was her haemoglobin postoperatively? Was any blood transfusion done?
R: The total estimated blood loss was 800 ml, her postoperative Hb was 9.3 g/dl. No transfusions needed
Details added at lines 98,99. “The total estimated blood loss was 800 ml, with a post operative Hb of 9.3 g/dl.”

Round 2
Reviewer 1 Report
Comments and Suggestions for Authors
The quality of Figure 5 is very poor. It is out of focus and the lesions stated are not visible (infarcts). The letters A and B are not on the same level and the figure lacks the scale bar. Please provide a better picture.
"Placental histopathological analysis" is the correct phrase, please keep it throughout the manuscript.
The other requests have been fulfilled.
Comments on the Quality of English LanguageMinor editing.
Author Response
dear reviewer thank you again for your precious suggestions
Figure 5 has been improved in quality as per your suggestion
the sentence has been changed as per your suggestion
best regards
